# Impact of COVID-19 on University Students: An Analysis of Its Influence on Psychological and Academic Factors [note 1]

**DOI:** 10.3390/ijerph191610433

**Published:** 2022-08-22

**Authors:** Gerardo Gómez-García, Magdalena Ramos-Navas-Parejo, Juan-Carlos de la Cruz-Campos, Carmen Rodríguez-Jiménez

**Affiliations:** 1Department of Didactics and School Organisation, Faculty of Education Science, Campus Universitario de Cartuja, University of Granada, 18071 Granada, Spain; 2Department of Didactics and School Organization, Faculty of Education and Sport Sciences, Campus Universitario de Melilla, University of Granada, 52071 Melilla, Spain

**Keywords:** COVID-19, life satisfaction, depression, academic motivation, higher education, structural equation modelling

## Abstract

The irruption of COVID-19 has had different consequences on mental health in the youth population. Specifically, the sector made up of university students has suffered an abrupt change of teaching modality because of the pandemic. As such, this paper aims to analyze the impact that COVID-19 has had on different personal factors of students: (i) satisfaction with life; (ii) lived uncertainty; (iii) depression, anxiety, and stress, as well as factors related to academic development; (iv) motivation and the creation of teaching and learning strategies during this period; and (v) the perception of the degree of adaptability to the new scenario brought about by the university system. For this purpose, a cross-sectional quantitative design was advocated through the elaboration of an SEM model, which included 1873 university students from Andalusian Universities (Spain). The results reflected the strong negative impact that the pandemic had, especially on the levels of life satisfaction and the indices of depression, anxiety, and stress of the students. Likewise, the findings reflected the relevance of the correct adaptability on the part of the university to these new circumstances. It is necessary for university institutions to focus their efforts on quality attention to students, in order to establish fluid communication with them and to adapt to their academic and personal needs.

## 1. Introduction

The World Health Organization declared in March 2020 that COVID-19 was a pandemic that affected all areas of society worldwide without exception. So much so that even today, different countries and societies are still fighting against it and establishing different measures in many areas to mitigate its effects and consequences. The rapid expansion of COVID-19 and all of the tremendously negative effects it has had on the health of the world’s population and on the economy at both the macro and micro levels has generated great levels of insecurity and stress in a large part of the population.

The pandemic brought with it major restrictions that persist to this day. These restrictions on mobility, the closure of non-essential businesses, and the limitation of social relationships have led to mental health consequences such as those noted above. In addition, unemployment rates have increased since the onset of the pandemic [1]. Job insecurity is an experience that has been associated with distress and negative feelings [2,3]. Throughout history, during pandemics or major crises, it has been shown that it is at these times that psychological problems in the population increase, as well as levels of anxiety, depression, and stress [4,5].

The extent of these problems is not yet fully known [6]. It is at present that studies are beginning to be carried out on these consequences in certain sectors of the population such as healthcare workers, analyzing the impacts of COVID-19 on their mental health—including insomnia, anxiety, and depression—and what other risk factors increase the probability of suffering from these diseases [7,8,9], or analyzing COVID-19 patients who also suffer from these mental disorders depending on certain demographic factors, such as older age, lower educational level or less physical activity [10].

Some population groups are more vulnerable than others to the psychosocial effects of pandemics [11]. Existing research focuses on the mental health problems of frontline medical personnel. The samples analyzed show that within this population group, it is women, nurses and those health professionals who diagnose, treat or provide nursing care to COVID patients who show more symptoms of distress, depression, stress or insomnia [12,13].

On the other hand, existing research also focuses on how individuals have lived with uncertainty throughout this period of the pandemic, and even today, they are still in this situation. Both the media and social networks have often contributed to this uncertainty due to the amount of information and data provided, which is an element that clearly influences people’s mental health [14]. The media can also provoke certain doubts among the population, along with changes in public attitudes and behavior. In contrast, sharing information in real time, with official data and analysis, will improve the ability of public health agencies and relevant stakeholders to respond to and understand the social dynamics of the rapidly increasing and changing dissemination of information and misinformation regarding the coronavirus and the outbreak and control measures. It will also reduce community panic and futile measures disproportionate to the cause [15].

Because of these processes, there is a more-than-palpable psychological burden on the population. A recently published study investigated the impact of the COVID-19 pandemic on the mental health burden of the Chinese public [16]. This study shows a high prevalence of generalized anxiety, depressive symptoms, and poor sleep quality due to stress during the COVID-19 pandemic. Thus, it is easy to conclude that this turbulent time in which we are immersed affects everyone’s mental health in some way or another, as all aspects of what was meant by everyday life have changed in some way or another.

## 2. Impact of COVID-19 on the Educational Scenario

The irruption of COVID-19 provoked, among many other things, the total closure of educational centers at all levels, meaning that teaching went from being face-to-face to online. The challenges that this has caused for students, having to deal with new, strange, and sometimes complicated situations, are remarkable for the consequences that have resulted from them.

Confinement, as an urgent measure to contain the virus, took place in March 2020, and involved an accelerated deployment of solutions for education to try to ensure academic continuity [17,18], along with a psychological impact for students and teachers [19].

Schools and universities were unprepared to deal with this unprecedented situation, which affected 1.57 billion students in 191 countries [20]. Therefore, they were faced with the challenge of dealing with the reorganization of their activities with immediacy and creativity, in order to avoid a negative outcome for students’ education [21]. In addition, they had a series of obstacles, such as low connectivity, a lack of online content, teachers who did not have the appropriate training for this type of distance learning led by ICT, and lack of technological resources and connectivity on the part of the students, among others [22]. Educational inequalities were particularly affected, as the exclusion of the most disadvantaged students increased, making it impossible for them to continue their education.

The compulsory and massive use of all kinds of ICT platforms and resources to guarantee the continuity of learning has been a great experiment in the implementation of educational technology. However, by being carried out in such an abrupt manner, these have generated unease among educational agents [23]. According to [24], university students were greatly harmed by the change from face-to-face to virtual classes, as communication with faculty was insufficient; they encountered connectivity problems, and were overloaded with homework and demotivated, which increased the level of anxiety and stress [25,26,27,28]. Teachers, took the situation as an opportunity to learn other didactic methodologies and, on the other hand, felt the stress of uncertainty and the workload. Resistance to change and the lack of digital competence of teachers have been the major impediments to distance education. Higher Education students feel that universities have not adapted to e-learning and assessments [29]. Because the measures adopted have not been sufficient, this has had an impact on a low level of learning and academic performance, affecting the average grades and jeopardizing students’ future employment.

Students have seen their motivation for learning diminished due to the fear of illness, the deprivation of liberty, difficulties in accessing the internet in some cases, and economic and family factors, etc. However, in most cases, they managed to achieve their academic goals [30,31]. Ref. [32] states that the effects of COVID-19 on university students have generated the following main difficulties: social isolation, anxiety, depression related to the situation generated by the pandemic, difficulty in adapting to ICT, maintaining a schedule and scenario to work continuously, economic aspects, accessibility to technological resources, and the poor planning of teachers [33,34,35,36]. A study carried out by [37] emphasized these negative aspects that students have experienced with the arrival of the pandemic, and that have negatively affected their life satisfaction in all of its dimensions, both personal and social. They conclude that for female university students, the emotional impact has been more acute; they have developed higher levels of fear and anxiety. These are aspects directly related to the difficulty of making decisions and looking to the future with optimism, generating situations that increase uncertainty [38].

After the cessation of the state of alarm, the result has been an increase in cases of anxiety and depression among university students due to the emotional changes produced by the disruption of daily routines and the reduction of physical contact [39]. Ref. [40] also reiterates that some of the negative consequences on students’ mental health are increased levels of anxiety, worry and fear, often leading to depression.

Students have generally taken the transition to virtuality negatively, which has led to behavioural and emotional changes that have affected wellbeing and academic performance [41]. Educational spaces have highlighted the satisfaction of learning needs, leading to the possibility of depressive symptoms [42].

Therefore, in terms of mental health, university students have been considered a vulnerable population, where resilience has played a fundamental role in maintaining emotional stability. Refs. [25,43] state that universities have the power to prevent and mitigate these negative effects on the mental health of their students.

According to [42], through self-determination theory, the causes influencing students’ motivation during this emergency situation have been related to a too-rapid transition between traditional face-to-face learning and digital learning. This theory details how socio-contextual factors affect student motivation through the satisfaction of their basic psychological needs: autonomy, competence, and affinity. When these three needs are met, students’ motivation to learn and engagement increase [44]. Autonomy refers to the self-regulation of particular actions and commitments. Competence is understood as the ability to effectively perform the given tasks. Moreover, affinity has to do with connection with others. Virtual or distance learning can satisfy the needs of autonomy and competence, but it does not attend to the need to relate, or to create this affinity with the rest of the educational agents, students, and teachers. During the pandemic, many students have felt isolated and disconnected [45].

In higher education, this situation caused by COVID-19 has increased anxiety levels due to the uncertainty generated in the labour market, especially in final year students [46], and has worsened the economic crisis and the income level of families [47]. Living in urban areas, having financial stability at home, living with family and having social support have been found to be protective factors against anxiety [48].

Women, on the other hand, have shown higher levels of anxiety than men [49]; however, men have stood out as suffering more depression than women [50]. In terms of age, higher measures of stress, anxiety and depression are found in younger students than in older ones [51].

## 3. Current Study and Research Objectives

This study also focuses on the configuration of a model (Figure 1) that investigates the relationship between the fear of COVID-19, life satisfaction, uncertainty, depression, anxiety and stress, motivation towards the learning process, and acceptance to the new technological scenario. Based on this idea, the following hypotheses are put forward:

**Hypothesis** **1** **(H1).**
*Fear of COVID-19 has a negative influence on levels of life satisfaction due to the possible influence on the perception of quality of life and the need to include changes in participants’ lives.*


**Hypothesis** **2** **(H2).**
*Fear of COVID-19 has a positive influence on uncertainty indices due to the possible influence of COVID-19 on participants’ insecurity.*


**Hypothesis** **3** **(H3).***Fear of COVID-19 has a positive influence on depression, anxiety and stress due to the possible influence on mental health states*.

**Hypothesis** **4** **(H4).**
*Fear of COVID-19 has a negative influence on motivation towards teaching–learning processes due to the possible decrease in academic performance.*


**Hypothesis** **5** **(H5).**
*Fear of COVID-19 has a negative influence on perceptions of the university’s adaptability to the new scenario due to participants’ facing unfamiliar academic scenarios during the pandemic.*


**Hypothesis** **6** **(H6).**
*Life satisfaction indices are positively associated with the adaptive conditions of the university system, as they allow for the successful completion of academic processes.*


**Hypothesis** **7** **(H7).**
*Life satisfaction scores are negatively related to depression, anxiety and stress scores because of their possible influence on participants’ health status.*


**Hypothesis** **8** **(H8).**
*Life satisfaction scores are negatively related to uncertainty because of the possible influence on the prospects of future academic grades.*


**Hypothesis** **9** **(H9).**
*Life satisfaction is negatively related to the motivation towards the new teaching–learning process because of the possible influence of the teaching–learning methods used during the pandemic.*


**Hypothesis** **10** **(H10).**
*Indices of experienced uncertainty are associated with levels of depression and anxiety and stress because of the possible influence of uncertainty on students’ mental health.*


**Hypothesis** **11** **(H11).**
*The uncertainty experienced is negatively related to the indexes of motivation towards the teaching–learning processes because of its possible relationship with the teaching–learning methods used during the pandemic.*


**Hypothesis** **12** **(H12).**
*The uncertainty indexes are inversely related to the perception of adaptability to the new training scenario proposed by the university due to its possible influence of the teaching–learning methods used during the pandemic.*


**Hypothesis** **13** **(H13).**
*Levels of depression, anxiety and experienced stress are negatively linked to motivation and the search for new learning strategies due to the possible influence of the students’ mental states on the teaching methods used during the pandemic.*


**Hypothesis** **14** **(H14).**
*The levels of depression, anxiety and stress experienced are negatively associated with the level of adaptability to the teaching promoted by the university because of the possible influence of students’ mental health status on teaching processes.*


**Hypothesis** **15** **(H15).**
*Indices of motivation and the search for new learning strategies are negatively related to the perception of adaptability to the new training scenario proposed by the university due to the possible influence that the pandemic exerted on motivation and attachment to teaching techniques during COVID-19.*


## 4. Methods

The use of a quantitative methodology was advocated to answer the research questions posed, and to satisfy the initially projected objectives [52].

### 4.1. Sample and Procedure

A cross-sectional study design was advocated by means of a self-administered survey in a sample of Spanish university students. Specifically, the study used a sample of 1873 students from the autonomous community of Andalusia, located in the southern part of Spain (*n* = 1873). The survey was disseminated via Google Forms through different official university channels of each of the Andalusian universities until a significant sample size was reached. The sampling criteria consisted of a convenience sample, which was sent to students enrolled in the eight universities of the single Andalusian district.

Likewise, the study participants responded to a series of questions referring to different sociodemographic variables, as well as to different standardised scales that determine the fear of COVID-19, current life satisfaction, depression, self-perceived stress and anxiety, uncertainty, and current academic motivation. Prior to completion, all of the participants gave informed consent. Likewise, all of the subjects were provided with information about the purpose of the research and the anonymous treatment of the data.

Table 1 presents the sociodemographic data of the participants. The sample consisted of 457 males and 1415 females, ranging in age from 17 to 59 (M = 22.42; SD = 4.451). This is a group of students in which most of their parents were working before the pandemic, and the effects of the pandemic caused many of them to lose their jobs. Likewise, a large percentage of the parents were on temporary lay-offs (ERTE, Expediente de Regulación Temporal de Empleo; Temporary Employment Adjustment Plan) during the period of the greatest incidence of the pandemic in the country. Regarding housing conditions while studying, the vast majority of students were living in a rented flat (58.8%), for which they had to terminate their contract due to the pandemic. As a result, most of them (80.9%) had to return to their family home (Table 2).

The study was applied at the end of June 2020. The State of Alarm protocol in Spain began on 14 March 2020, and from that date until 9 May there was house confinement in Andalusia. From this date, the de-escalation process took place in the country. In addition, this was the time when the final evaluation period of the university course was taking place; as such, we considered it appropriate to carry out the study in this period, as it would be a point of reflection for students on this atypical academic year in which they could thoroughly assess the course, and the possible impacts caused.

The study was conducted according to the guidelines of the Declaration of Helsinki, and was approved by the Ethics Committee of the University of Granada (REF:0706/2020).

### 4.2. Measures

The scales are standardized, and can be accessed through the citations provided in the description of each of the constructs.

#### 4.2.1. COVID-19 Fear Scale

The COVID-19 Fear Scale was developed by [53]. It is composed of 7 items that follow the 5-point Likert format (1 = strongly disagree; 5 = strongly agree). It is responsible for measuring the indices of fear and anxiety produced in people because of COVID-19. The Cronbach’s alpha coefficient obtained in this study was 0.929.

#### 4.2.2. The Life Satisfaction Scale

The life satisfaction scale was developed by [54]. The scale has five items in the form of a 7-point Likert scale (1 = Not at all agree; 7 = Strongly agree). The scale has only one dimension. It is intended to analyse the students’ levels of life satisfaction during this complex period. The Cronbach’s alpha coefficient obtained in this study was 0.936.

#### 4.2.3. Intolerance of Uncertainly Scale (IUS-12)

The Intolerance of Uncertainly Scale is an abbreviated version of the IUS [55] developed by [56], and is based on 12 items with a 5-point Likert scale (1 = not at all; 5 = very much). It measures intolerance to uncertainty as the tendency to react negatively on an emotional, cognitive, and behavioural level to uncertain situations and events. It is configured around two dimensions: prospective anxiety, which measures fear and anxiety related to the future, and inhibitory anxiety, which measures the inhibitory action or experience of uncertainty. The Cronbach’s alpha coefficient obtained in this study was 0.934.

#### 4.2.4. Depression Anxiety Stress Scale-21 (DASS-21)

The DASS-21 scale, developed by Lovibond and recommended by the Australian Psychological Association, consists of three self-report scales designed to measure the emotional states of depression, anxiety and stress [57]. Each of the three scales contains 7 items, divided into subscales of similar content. The scale is a 4-point Likert scale (0 = has not happened to me; 3 = has happened to me a lot, or most of the time). This instrument is used to determine the level of specified negative emotional states. In the current research, the use of its validated version in the Spanish language was advocated [58]. The purpose of its application is to indicate the psychological state of university students during the pandemic. The Cronbach’s alpha coefficient obtained in this study was 0.902.

#### 4.2.5. Motivated Strategies for Learning Questionnaire (MLSQ-SF)

The Motivational Strategies for Learning Questionnaire–Short Form (MSLQ-SF) was developed by [59]. It consists of 40 questions grouped around three dimensions: (i) Motivation scale, items linked to task evaluation and task anxiety; (ii) Learning strategies, made up of developmental strategies, critical thinking, and self-regulation of learning; and (iii) Resource management strategies, structured according to time and study habits, self-regulatory effort, and intrinsic goal-setting orientation. The scale is Likert type 5 (1 = never; 5 = always). This scale was chosen to find out the levels of motivation, as well as the students’ strategies to promote learning in the new scenario during the home confinement and the academic course that took place virtually. The Cronbach’s alpha coefficient obtained in this study was 0.948.

#### 4.2.6. Unified Theory of Acceptance and Use of Technology (UTAUT)—Facilitating Conditions (FC)

Facilitating conditions are defined as the “degree to which an individual believes that an organizational and technical infrastructure exists to support use of the system” [60]. This construct is considered to be one of the strongest predictors in the behavioural intention to use a technology [61], and presents a significant influence [62]. In the educational context, facilitating conditions refers to human, organizational, and technical support for using technology [63]. This construct is measured through a 7-point Likert scale. This dimension was added with the aim of analysing the degree of adaptation that students perceive that universities have had when adapting to the virtual mode of teaching, and what their degree of acceptance is. The Cronbach’s alpha coefficient obtained in this study was 0.907.

#### 4.2.7. Statistical Analyses

Firstly, descriptive statistics were calculated to evaluate the self-perception of the students in terms of their expressed fear of COVID-19, levels of depression, satisfaction with life, and uncertainty experienced so far.

Finally, a structural equation model (SEM) was set up to establish the correlation coefficients between the measured constructs.

The data were analysed using the statistical programs IBM SPSS version 25 and IBM SPSS Amos, version 24.

## 5. Results

The descriptive statistics shown in Table 2 reflect indices of fear of COVID close to the average, as well as levels of life satisfaction in students which were slightly above the standard. With respect to the perceived level of uncertainty, this is slightly above average, along with the levels of motivation and learning strategies. The high values for the depression, anxiety, and stress constructs, as well as the facilitating conditions, stand out.

On the other hand, the level of variability among the responses is high, as indicated by the standard deviation coefficients, such that it is likely that there are outliers in the data distribution. This statement is confirmed when visualizing the coefficients of asymmetry and kurtosis, which are mostly negative, which determines that most of the participants’ answers are below the arithmetic mean indicated.

Subsequently, structural equation modelling (SEM) was used to check whether the hypothesized model (Figure 1) coincided with the data collected. The correlations between the different constructs are presented in Table 3, with all of them being statistically significant (*p* < 0.001). The coefficients obtained highlighted moderate–strong relationships between the constructs. In the first place, the strong positive relationships between SAT and FC, COVID and UNC, and COVID and DAS exceed more than half of the dependence index. On the other hand, negative correlation relationships were also elucidated, such as those found between COVID and FC, COVID and SAT, SAT and UNC, and SAT and DAS, for which the coefficients indicate a medium non-linear dependence relationship.

After this, the SEM model was set up. In it, relationships of influence between the different latent variables were observed. Associations between variables such as FC and SAT, DAS, and UNC, and COVID and UNC stand out for the strong linearity links they present. In contrast, non-linear relationships were found, as was the case of COVID and FC, COVID and SAT, and SAT and UNC. Regarding the goodness-of-fit indices, the results showed a relatively good fit between our model and the data (χ^2^/df = 1.761, *p* = 0.000; RMSEA = 0.0079; NFI = 0.959 CFI = 0.976; GFI = 0.967; AGFI = 0.951). If we look at the configured model (Figure 2), relationships were observed. All of the relationships obtained were statistically significant.

## 6. Discussion

The evidence presented in this paper reflects the clear impact that COVID-19 has had on the participating university students, a sector of the population that has been greatly affected in terms of mental health by the pandemic [51]. In this study, we examined the impact that this has caused (COVID-19) in different personal factors: life satisfaction (SAT), depression, anxiety, stress (DAS), and uncertainty (UNC); as well as factors with respect to their training: the motivation and configuration of teaching–learning strategies (MOT), and perceptions regarding the degree of adaptability of the university to the new scenario that has occurred (FC). Likewise, a structural model was presented that elucidated the unobserved interactions between the different personal and academic factors that may have affected university students during this stage.

The results of the study indicated that the students presented an average degree of fear towards COVID-19 and a high value of uncertainty, while high levels of depression, anxiety and stress were observed during this period. This is in line with different studies that conclude that COVID-19 mainly has a strong impact on stress and anxiety indexes [7,16], as well as depressive disorders of different natures [8]. However, and in the face of this situation, the students determined to a degree that we can consider appropriate that they found new teaching–learning strategies, as well as that their motivation towards their formative process prevailed. On the other hand, although it is true that high results were obtained in terms of the facilitating conditions proposed by the university, as well as a value close to the average in life satisfaction, we cannot consider it transcendental due to the high degree of variability found in the answers. In any case, we could affirm that, despite the high coefficients for these constructs, they have not had a positive effect on the indices of depression, anxiety, and stress of the participating university students, in contrast to what has been affirmed by studies in this sense [25]. This could be due to the notorious economic impact that students claim that COVID-19 has triggered in their lives, either because of their own work situation or that suffered by their families, which is a conclusion that coincides with that obtained by [47].

In addition, the significance of the structural model obtained invites us to make a series of reflections regarding the different findings that it presents, and the hypotheses raised at the beginning of the research: the adverse relationship between the fear of COVID and life satisfaction (H1), the growth of uncertainty (H2), or the statistically strong significant association between the fear of COVID-19 and the growth of stress, anxiety, and depression (H3). Regarding the first statement, it is obvious to link the occurrence of the disease with the decrease in the quality of life of students [9]. In the same way, the growth of uncertainty, the misinformative phenomena surrounding the virus, its nature, its effects, and possible ways of resolution, along with the possible consequences on their future education and work, promote a greater sense of panic towards the virus and its phenomenology. In relation to this evidence, it would also be related to the weight of uncertainty in the life satisfaction of students (H8), which is in line with previous studies [14]. Likewise, there is the incidence of uncertainty in anxiety, stress, and depression (H10), to which we could add the feeling of panic and dissatisfaction (H7), as stated in the scientific literature [15]. In sum, the uncertainty that existed during the change of educational scenario, and the accelerated solutions that the educational system had to implement at the beginning and during the confinement had a negative impact on the academic motivation indexes (H11), which is a statement that we share with previous works that turn in this sense [17,18].

While exposure to COVID-19 in general could lead to a decrease in positive perceptions of the academic process (H4), the results of the present study found a strong statistically significant association between the proliferation of fear of the virus and the pursuit of new teaching and learning strategies, the development of self-regulation towards learning, and an increase in motivation to achieve academic goals; this is a result which is in line with [44]. Likewise, preserving this academic motivation turned out to be a clear indicator for reducing depression, anxiety, and stress (H13). In order to promote this, we consider it necessary that the teacher communicates fluidly with students, provides continuous feedback on academic progress, and prevents the student from having the perception of isolation while following the virtual modality [45]. However, the adverse relationship found between motivation and the search for new strategies and the indices of life satisfaction is surprising, being the opposite of the hypothesis proposed at the beginning of the research (H9).

The majority presence of women (75.6%) in the study is consistent with their greater presence in degrees such as Educational Sciences or Psychopedagogy, the percentages of which can be over 90% [64]. In general, women mainly choose degrees in the Humanities, Experimental Sciences, Social Sciences, Law and Health because they like it, for their vocation, or to help other people [65]. This greater representation of women is not a limitation but a mirror of the situation of universities in Spain and the increase in female representation in higher education. Future studies would need to determine the mental health status prior to the COVID-19 pandemic, but this retrospective study seems very difficult, if not impossible, to carry out on the same sample. In the pre-pandemic state, university women have shown higher scores on depression and anxiety tests [66] although the interrelation of eating disorders and depression, which are more frequent in university women, should be taken into account in the future [67].

A lack of communication between teachers and students may be linked to the need to maintain social ties with others and participate in social activities, which, without them, may have led to the development of a sense of insecurity during the pandemic. During this time, other populations had difficulty accessing basic goods and services, and a greater number of mental health problems developed [68]. We speculate that the different containment policies between countries, greater access to online services, and different demands and expectations may have reduced mental health problems in our sample.

Finally, COVID-19 has had a negative repercussion on the students’ perception of the abrupt change that has taken place in the educational process (H5), which is an idea already reflected by other authors [23]. Likewise, the perception of adaptability to the new educational scenario that university students have is a strong dependency factor in the promotion of higher rates of life satisfaction (H6). It is a key factor, the development of which implies a lower index of stress, anxiety, and depression in the student body (H14), decreases uncertainty (H12), and creates a greater motivation towards academic progress (H15). Knowing the guidelines according to which distance learning is going to take place, the possible methods of evaluation during this period or mastering the technological resources with which the training process takes place are some of the possible indicators that promote an improvement in the personal factors of students, and it is necessary for universities to work on them [22].

## 7. Limitations and Future Research Directions

The main limitation observed is that the study followed a cross-sectional design and, therefore, the results are limited to the specific moment in which the data were collected, rather than representing the entire period in which the students were subjected to quarantine or confinement as a much longer period in which the change of teaching modality took place. Likewise, the results are limited to the specific sample studied, such that they cannot be generalized to the entire Spanish university population, as they only describe the moment of that reality, in this case, of the students enrolled at Andalusian Universities.

On the other hand, in terms of future lines of research, our work opens up the possibility of carrying out studies that deal with predictors of the different constructs we are talking about, as well as studies that propose more factors linked to the mental health of university students. These studies could address the medium- to long-term consequences of the pandemic if they are carried out in the post-pandemic period, and may offer the option of making comparisons with other universities in Spain, or even other countries.

## 8. Conclusions

The predictive power of the SEM model designed is sensitive to the determination of the influences between the different variables that compose it: fear of COVID-19, life satisfaction, uncertainty, anxiety and stress, motivation and configuration of new teaching–learning strategies, and perceptions of the degree of adaptability of the university to the new scenario, observing positive and negative relationships of influence between the different latent variables.

Both the pandemic and the confinement measures have had a strong negative impact on students’ levels of life satisfaction, depression, anxiety and stress. In order to face these challenges, students and universities themselves have shown great adaptability to these new circumstances by promoting personal protection measures and academic adaptations, but it is still necessary that university institutions continue to strengthen the attention to the student body, establishing fluid communications with them in order to adapt to emerging academic and personal needs.

Our study indicates that the outbreak of COVID-19 has had an influence on the mental health of university students, who have undergone an abrupt change in teaching methods because of the pandemic. Because the mental health of students is related to their academic performance, and even to university dropout, it is necessary that academic institutions themselves face measures that attempt to mitigate this influence, and that they carry out pragmatic studies to evaluate the effectiveness of these interventions in reducing the deterioration of academic function.

The results suggest that mental health problems may represent a much higher proportion than expected, and may lead to a deterioration of the academic role among university students. The incidence of mental health problems derived from the pandemic could emerge in the coming years with greater intensity, such that it is necessary to establish both academic and institutional anticipatory guidelines, including the recognition of mental health problems, the requests for mental healthcare by the student body, the social awareness of mental health problems, and the identification of mental health problems that may underlie poor academic performance.

As colleges and universities begin to grapple with the growing recognition of the important role that mental health plays in student success, especially in relation to the effects of the pandemic, more attention will need to be paid to broad-based interventions designed to improve the mental health of the student body. Given the magnitude of the problem, multidisciplinary solutions, more research to evaluate the effects of innovative and scalable mental health interventions, and the development of methods to triage college students in need of treatment are needed.

However, and despite having lived through a formative period that we can describe as extraordinary in nature, we cannot ignore the need for pedagogical teaching strategies and digital training to promote quality teaching. This emergency situation has highlighted the need for the future of transforming traditional education systems, the importance of promoting an immersion of technological resources in the ordinary teaching of students, and the promotion of fluid and continuous communication with students.

## Figures and Tables

**Figure 1 ijerph-19-10433-f001:**
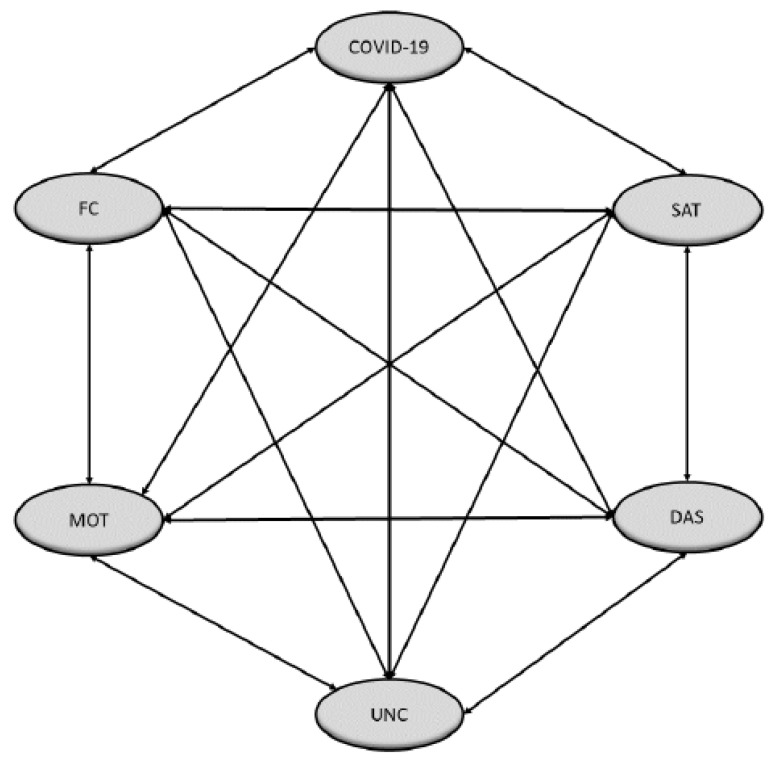
Theoretical model to determine the level of dependence between latent variables. Note: Own production.

**Figure 2 ijerph-19-10433-f002:**
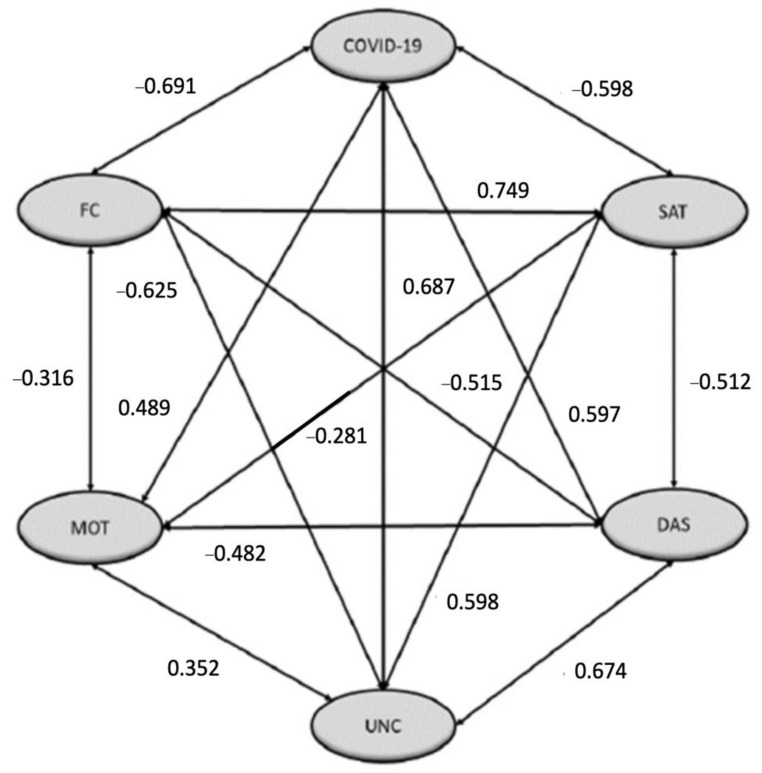
Estimations of the structural equation model. The relations were significant at *p* < 0.001. Note: Own production.

**Table 1 ijerph-19-10433-t001:** Gender and age.

Variable	*n*	%
**Gender**		
Men	457	24.4
Women	1415	75.6
**Age**		
Between 17–30 years old	1174	94.7
Between 31–50 years old	94	4.9
Over 51 years old	3	0.4

Note: Own production.

**Table 2 ijerph-19-10433-t002:** Descriptive statistics.

**Construct**	**Mean**	**SD**	**Asymmetry**	**Kurtosis**
COVID	2.80	1.423	0.270	−0.876
SAT	4.71	1.860	−0.613	−0.567
UNC	3.37	1.135	−0.216	−0.914
DAS	1.98	0.986	−0.543	−0.694
MOT	3.84	1.151	−0.682	−0.128
FC	5.14	1.958	−0.956	−0.154

Note: Own production.

**Table 3 ijerph-19-10433-t003:** Covariance relationships between the constructs.

H	Relation			Estimate	SE	C.R.	*p*	Decision
H1	COVID	<-->	SAT	−0.767	0.046	−18.521	***	Accepted
H2	COVID	<-->	UNC	0.621	0.045	19.152	***	Accepted
H3	COVID	<-->	DAS	0.599	0.040	18.564	***	Accepted
H4	COVID	<-->	MOT	0.355	0.034	15.565	***	Rejected
H5	COVID	<-->	FC	−0.921	0.072	−20.956	***	Accepted
H6	SAT	<-->	FC	0.852	0.097	22.473	***	Accepted
H7	SAT	<-->	DAS	−0.597	0.029	−16.720	***	Accepted
H8	SAT	<-->	UNC	−0.687	0.052	−17.458	***	Accepted
H9	SAT	<-->	MOT	−0.199	0.019	−11.241	***	Accepted
H10	UNC	<-->	DAS	0.414	0.037	18.246	***	Accepted
H11	UNC	<-->	MOT	0.245	0.030	13.351	***	Rejected
H12	UNC	<-->	FC	−0.812	0.054	−18.32	***	Accepted
H13	DAS	<-->	MOT	−454	0.032	−10.548	***	Accepted
H14	DAS	<-->	FC	−0.648	0.054	−14.956	***	Accepted
H15	MOT	<-->	FC	−0.424	0.033	−13.453	***	Accepted

Note: Own production. *** denotes significance at *p* < 0.001.

## Data Availability

The datasets generated and analyzed during the current study are available from G.G.-G. on reasonable request.

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
