# Peer review of "Impact of COVID-19 on University Students: An Analysis of Its Influence on Psychological and Academic Factors†"

_ijerph, 2022, doi:10.3390/ijerph191610433_

Round 1

Reviewer 1 Report

Thank you for the revisions.

Author Response

Thank you very much.

We hope that this research will be a significant contribution to the existing literature on the subject.

Reviewer 2 Report

It reads better now. My only concern is that most of the goodness-of-fit indices indicate the model perfectly fits the data. This is too good to be true. I have never seen RMSEA = .0000, NFI=.999, CFI = .999, GFI= 1 in any previous study. This is especially questionable when I compare it with the previous version of this manuscript. All the estimates in Table 3 and Figure 2 are exactly the same as that in the previous manuscript. Then how could it be possible that the goodness-of-fit indices got such a huge improvement?

I would like to believe that the authors have mistakenly copied the previous figures and tables from the previous manuscript. So please check and fix it.

Author Response

Dear reviewer,

Thank you very much again for your thorough review of our manuscript. Your feedback certainly helps us to improve both the quality of the manuscript and on a personal level as researchers.

We hope that with these latest changes we have met your expectations and addressed your concerns.

First of all, we apologise that we have not changed table 3 and figure 2, they have been updated.

Likewise, we have again reviewed and adjusted the goodness-of-fit indices, which we believe are now properly adjusted.

We thank you again for your timely comments and thank you for giving us the opportunity to improve our work.

Yours sincerely.

Round 2

Reviewer 2 Report

I think the present manuscript is now publishable. Thanks for addressing my previous concerns. 

This manuscript is a resubmission of an earlier submission. The following is a list of the peer review reports and author responses from that submission.

Round 1

Reviewer 1 Report

Thanks for inviting me to review this paper. I think the topic is interesting but the current manuscript needs to be carefully revised (some section may need to be rewritten) in order to be publishable. Please see my detailed comments below.

Major:

1.      The introduction section should provide the background info and lead the argument to what the authors is going to discuss in this paper. However, this introduction failed to raise the question of this paper. Even worse, by reading this section, it is still confusing how this paper will contribute to the literature. I suggest, if possible, completely rewrite this section. You may start with a brief introduction of the pandemic and its corresponding mental health consequences in the first paragraph and then narrow down to what you are going to talk about in this paper, explicitly.

2.      The conceptual framework looks interesting but was not clearly explained. Please specify EACH correlation (hypotheses) in the previous section. EACH one of them.

3.      Section 4.2. Please provide the questionnaire with all the items in an appendix. I ask for this because it seems that this study synthesises more than five psychological scales, most of which has some twenty items. This will make the survey an over-100-items one, which will require about 20-40 minutes to finish. It is very doubtful how could the authors approach so many students within one single university who were willing to complete such an extremely long survey. So, provide the full list of items to increase the transparency of your paper.

4.      According to the goodness of fit indices, the model is poorly fitted. χ2/df should be at least <5, CFI, TLI, and NFI should be>0.9. and please provide the value of AGFI instead of GFI or add AGFI because it is a better index. The model is poorly fitted probably because you did not do the EFA before the structural model. Please follow this procedure (Koufteros, 1999).

5.      In section 4.2, I think the long descriptions of the existing scales you chose could be deleted. You could simply say which scales you have used (and the reference). It would be clear enough.

6.      The discussion of this paper could go deeper. Currently this discussion section is merely summarising the results. This is not even a discussion. I suggest to rewrite this section. You could try to locate your study in a wider context. For example, you could compare university students with other population groups who may suffer from other forms of mental health problems (Liu et al., 2022). What makes it different? Is that because of different containment policies? different access to online services? different demands and expectations?

Minor:

1.      There are quite a few typos, please carefully check the content (for example in Table 1, “Genre” should be “gender”, “man” should be “men”, “over de 51 anos” should be “over 51 years old”)

2.      A proofreading service may be necessary since quite a few sentences are very difficult to understand because of the English.

3.      In table 1, there is a question about “To what extent has COVID-19 affected you at work?” I don’t get this. Are they students? Why did you ask about their working experience?

4.      In fact, I don’t know what the lower part of Table 1 is for. It is quite confusing.

5.      Line 238, what do you mean by “house arrest”? It is really confusing.

6.      Female respondents were two times more than male respondents. Is this a problem? You could easily solve this by providing the statistics of the university. If the gender ratio of the university is considerably lower than this, it might be a problem and you may add it as a limitation.

Reference

Koufteros, X. A. (1999). Testing a model of pull production: A paradigm for manufacturing research using structural equation modeling. Journal of Operations Management, 17 (4), 467–488.

Liu, Q., Liu, Z., Lin, S., & Zhao, P. (2022). Perceived accessibility and mental health consequences of COVID-19 containment policies. Journal of Transport & Health, 101354.

Reviewer 2 Report

This is a promising paper on the much needed topic that we are only beginning to understand. Please see my comments that are aimed at clarifying your points:

line 131-132 incomplete sentence

line 142-144 unclear sentence

line 146 "women have shown higher levels than men" --> levels of what?

Research objective is unclear. The sentence begins with "furthermore," which means "in addition to something." That "something" is missing.

Since you stated the hypotheses, please identify the variables. Table 1 shows survey questions, not variables (apart from gender and age).

Did you have any sampling criteria? Why was this particular university chosen? How did you arrive at the N? What was the response rate?

Please be more precise about specific methods/ techniques used. The statement "A wide range of statistical tests were used" is too vague. This does not look like an evaluation study, so descriptive stats were used to determine .... and SEM was employed to examine ... please be precise rather than saying "to determine the possible causal relationships between the different constructs mentioned above, establishing the
correlation coefficients between them."

Line 418 "direct relationship" -- this was not a randomized experiment. You mean strong, statistically significant association between ...

lines 432-434 "Likewise, regarding the perception of the university's adaptability --> whose perception?  to the new educational scenario, it --> What does "it" refer to? is configured as a strong dependency factor to promote higher rates of life satisfaction (H6). It is a key factor..."

Lines 450-453 "On the other hand, in terms of future lines of research, our work opens the possibility of carrying out studies that deal with predictors of the different constructs we are talking about, as well as studies that propose more factors linked to the mental health of university students." --> Please expand and explain, using precise language.

Conclusions: "The predictive power of the model designed..." Which model?

Thank you.